# Low maternal vitamin D is associated with increased risk of congenital and peri/postnatal transmission of Cytomegalovirus in women with HIV

Allison Bearden[1]*, Kristi Van Winden[2¤], Toni Frederick[1], Naoko Kono[3], Eva Operskalski[1], Raj Pandian[4], Lorayne Barton[5], Alice Stek[2], Andrea Kovacs[1]

1 Department of Pediatrics, Division of Infectious Diseases, Maternal, Child, Adolescent/Adult Center for Infectious Diseases and Virology, Keck School of Medicine of USC, University of Southern California, Los Angeles, CA, United States of America, 2 Department of Obstetrics and Gynecology, Division of Maternal and Fetal Medicine, and Maternal, Child, Adolescent/Adult Center for Infectious Diseases and Virology, Keck School of Medicine of USC, University of Southern California, Los Angeles, CA, United States of America, 3 Department of Preventive Medicine, Keck School of Medicine of USC, University of Southern California, Los Angeles, CA, United States of America, 4 Pan Laboratories, Irvine, CA, United States of America, 5 Department of Pediatrics, Division of Neonatology, Keck School of Medicine of USC, University of Southern California, Los Angeles, CA, United States of America

¤ Current address: Department of Obstetrics/Gynecology, Division of Maternal-Fetal Medicine, Kaiser Permanente Northern California Region, Oakland, CA, United States of America
* allison.bearden@usc.edu

**Data Availability Statement:** We are unable to publicly share the dataset from this study due to ethical and legal restrictions. According the

## Abstract

### Background

CMV infection of the fetus or neonate can lead to devastating disease, and there are no effective prevention strategies to date. Vitamin D is a potent immunomodulator, supports antiviral immune responses, and plays an important role in placental immunity.

### Methods

Retrospective cohort study to evaluate the impact of low maternal vitamin D on congenital and early postnatal transmission of CMV among HIV-infected, non-breastfeeding women and their HIV exposed but negative infants from an urban HIV clinic. Vitamin D panel was performed on stored maternal plasma obtained near time of delivery. Infant CMV testing at 0–6 months included urine and oral cultures, and/or serum polymerase chain reaction testing.

### Results

Cohort included 340 mother-infant pairs (births 1991–2014). Among 38 infants (11%) with a CMV+ test between 0–6 months, 4.7% (14/300) had congenital CMV transmission (CMV+ test 0–3 weeks), and 7.6% (24/315) had peri/postnatal CMV (CMV+ test >3 weeks-6 months). Women with lower calcitriol (1,25-dihydroxyvitamin D), the active form of vitamin D, were more likely to have an infant with congenital (OR 12.2 [95% CI 1.61–92.2] $P$ = 0.02)

University of Southern California's Office of Ethics and Compliance, there is no way to make the dataset truly de-identified. The dataset is made up solely of women with a diagnosis of HIV and their offspring. The diagnosis of HIV is an indirect identifier and considered sensitive information requiring additional protections. Additional indirect identifiers in the dataset include age, race/ethnicity, and sex. Additionally, the consent signed by the study participants did not include a provision for their data to be made publicly accessible. Data requests can be directed to department administrator, Carlota Obnillas, obnillas@usc.edu, and if the request is approved, the USC Stevens Center for Innovation will assist and create a Data Transfer Agreement.

**Funding:** This work was supported by the SC CTSI (https://sc-ctsi.org), (NIH/NCRR/NCATS) through Grant UL1TR000130, awarded to AK and AB. Its contents are solely the responsibility of the authors and do not necessarily represent the official views of the National Institutes of Health. This study was funded in part by a seed grant provided by the Department of Obstetrics and Gynecology at the University of Southern California awarded to KV. RP received support in the form of a salary from Pan Laboratories, Irvine, CA. The funders had no role in study design, data collection and analysis, decision to publish, or preparation of the manuscript. The specific roles of these authors are articulated in the 'author contributions' section.

**Competing interests:** RP is the salaried Director of Pan Laboratories, Irvine, CA. There are no patents, products in development or marketed products to declare. This does not alter our adherence to PLOS ONE policies on sharing data and materials.

and peri/postnatal (OR 9.84 [95% CI 2.63–36.8] $P$ = 0.0007) infections in multivariate analyses, independent of maternal HIV viral load and CD4 count.

## Conclusion

This study demonstrates an association between inadequate maternal calcitriol during pregnancy and increased congenital and early postnatal acquisition of CMV among non-breastfeeding women with HIV and their HIV negative infants.

## Introduction

Human Cytomegalovirus (CMV) infection is the most common congenital infection worldwide. It is the top non-genetic cause of childhood deafness in the world and can lead to neurologic and neurodevelopmental disorders, multisystem illness, growth and development abnormalities, and death.[1,2] Approximately 50–70% of women of childbearing age in developed countries are CMV infected, with the highest prevalence among women of lower socioeconomic status.[3] Seroprevalence approaches 100% among women of child-bearing age in resource-limited countries and in those with Human Immunodeficiency Virus (HIV) infection.[2,4] Mother-to-child-transmission (MTCT) of CMV can occur prenatally (congenital infection), during birth, and postnatally through breast milk.[5] Mothers and other caregivers can also transmit CMV to their infants postnatally through infected secretions.[2] Maternal CMV infections and reactivations are often asymptomatic and unnoticed, and unlike HIV, there are currently no effective strategies widely implemented for the prevention of MTCT of CMV.[6] Rates of congenital CMV are often higher among infants of women with HIV infection, making them an ideal population for study.[2,4,7–11]

Vitamin D is obtained either from exposure to ultraviolet light or from the diet. In addition to its role in calcium homeostasis and skeletal health, Vitamin D is a well-known and potent modulator of the immune system.[12] Vitamin D supports immune system antiviral responses through the induction of autophagy and production of antimicrobial peptides like cathelicidin, and likely plays an important role in helping to protect the developing fetus from infections during pregnancy.[13–16] A multitude of cells in the body have the vitamin D receptor and many cells, including the cells of the placenta, also have the ability to convert 25-hydroxyvitamin D (25(OH)D), the main circulating form of vitamin D, to its bioactive form, 1,25-dihydroxyvitamin D (1,25(OH)D$_2$).[17,18] This allows for local production of 1,25(OH)D$_2$ and the subsequent vitamin D-dependent antimicrobial immune responses in the setting of specific conditions or stimuli.[15,18–20]

Vitamin D's important role in supporting the immune system's antiviral functions, including those at the level of the placenta, suggests its relevance to MTCT of CMV in utero. Additionally, vitamin D may contribute to the immune system's ability to limit viral shedding and therefore play a role in limiting perinatal and early postnatal CMV transmission. In order to explore these hypotheses, we conducted a retrospective study, nested within a longitudinal prospective cohort study, evaluating the impact of low maternal vitamin D on congenital and peri/postnatal acquisition of CMV among HIV-infected, non-breastfeeding women and their HIV exposed but negative infants born between 1988 through 2015 at the Maternal, Child and Adolescent/Adult Center for Infectious Diseases and Virology (MCA) at the LAC+USC Medical Center, in Los Angeles, California.

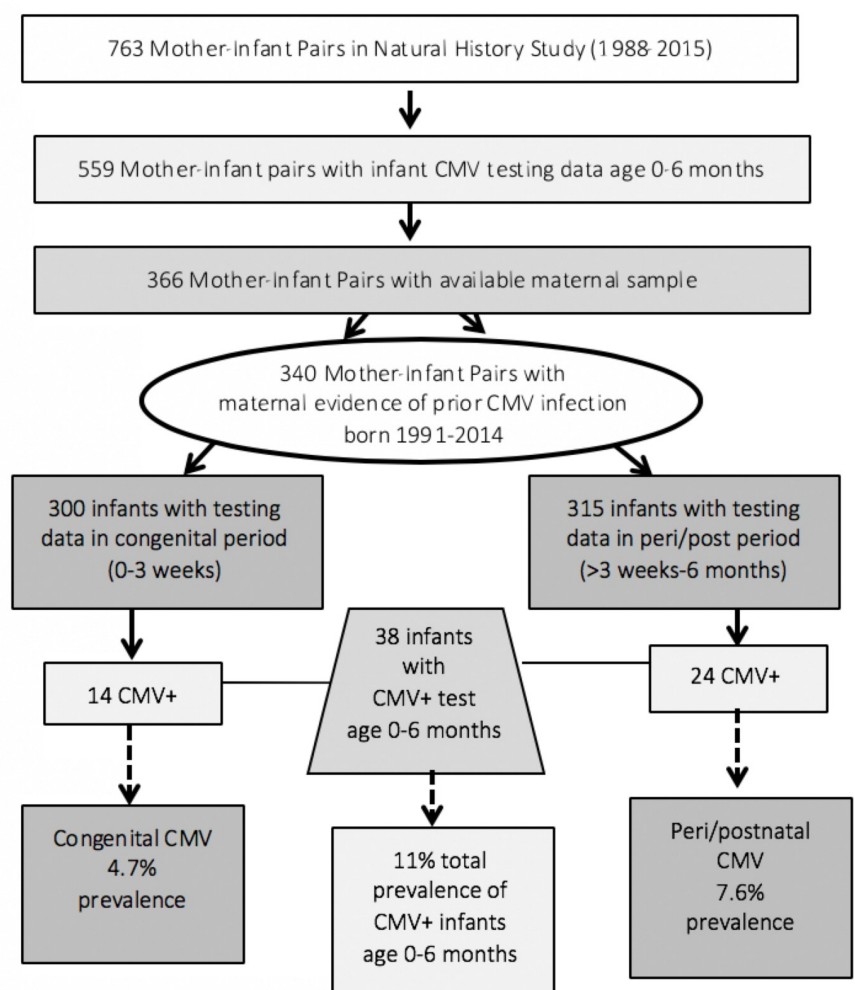

**Fig 1. Study cohort based on infant CMV testing results.**

## Methods

### Study design and participants

MCA is a comprehensive HIV clinic, serving women and their families. It is Los Angeles County's largest referral site for HIV-infected pregnant women and their children, and cares for those who are under or uninsured. Informed consent was obtained for mothers and their newborns receiving care at MCA to participate in the University of Southern California Health Sciences Institutional Review Board–approved Natural History Study. The cohort design and participant selection for the current study are summarized in Fig 1. Mother-infant pairs were eligible for inclusion in this study if 1) they were both enrolled in MCA's Natural History Study, 2) the mother was HIV-infected with evidence of CMV infection prior to the birth of the child, 3) the infant was HIV uninfected and had CMV testing between the ages of 0 to 6 months, and 4) stored maternal plasma obtained during pregnancy was available for vitamin D analysis. Among 559 mother-infant pairs with infant CMV testing between birth and age 6 months, 366 mothers had stored plasma available for vitamin D testing. Among these women, 340 had evidence of CMV infection: 312 were CMV seropositive and 28 with missing CMV

results, had infants who were CMV IgG+ at or near the time of birth. This was considered to be a transfer of maternal antibody and thus these mothers were considered CMV+ and included in the cohort.

CMV testing in HIV-exposed infants was done as part of MCA's routine care or as part of other MCA research studies. This testing included culture of urine and oral swabs, and polymerase chain reaction (PCR) studies of blood. Cultures were performed using standard virologic methods of either shell-vial or tube cultures. PCR was performed by contracted send out laboratories (Quest Diagnostics, Focus Diagnostics) or in some cases, PCR was performed for other research studies by the MCA Laboratory. CMV tests available for analysis in this study included 1,101 urine cultures from 312 infants, 478 oral cultures from 158 infants, and 67 CMV blood PCR results from 55 infants. Congenital CMV was defined as 1 or more positive CMV tests between the ages of birth to 21 days. Peri/postnatal infection was defined as 1 or more positive results between 22 days to 6 months without a positive test during the congenital period.

## Vitamin D testing

Stored maternal plasma samples were used for evaluation of vitamin D. Vitamin D is very stable and samples may be frozen indefinitely and can withstand several freeze-thaw cycles without impacting results.[21] Specimens obtained at the time closest to the infant's birth were used, with 50% obtained on the day of birth, 40% within 7 days of birth, 6.5% in the third trimester but before 7 days of birth, 3% in the second trimester, and 0.5% in the first trimester. Plasma samples were collected and stored in tubes containing either Ethylenediaminetetraacetic acid (EDTA), heparin, or Acid Citrate Dextrose (ACD). Samples were sent to Pan Laboratories, Irvine, CA for Vitamin D testing. Measurement of total 25(OH)D was performed by immunoassay (IDS ELISA, AC-57) using kits from Immuno-Diagnostic Systems (IDS), Phoenix, per package insert. The assay has a cross reactivity of 75% with 25 hydroxyvitamin D2. The assay has a sensitivity of 2.5 ng/ml and total interassay variation <8.7%.[22] Total calcitriol $(1,25(OH)D_2)$ was measured by immunoassay (IDS ELISA, AC-62), using kits from IDS. Calcitriol $(1,25(OH)D_2)$ was first purified by immuno-affinity purification and immunoassay was performed to quantitate calcitriol. The assay has a sensitivity of 2.5 pg/ml and total interassay variation is <15%.[23] Sufficient 25(OH)D levels were defined as ≥32 ng/mL, 80 nmol/L, insufficient as 21–31 ng/mL, 52.5–77 nmol/L, inadequate as 11–20 ng/mL, 27.5–50 nmol/L, and deficient as ≤10 ng/mL, 25 nmol/L.

## Statistical analysis

Demographic and HIV related variables evaluated included maternal age, race, ethnicity, HIV viral load categorized as <400 or ≥400 copies per ml, CD4 cell count categorized as <200 or ≥200 cells/mm³, mode of delivery, infant gender, season, and antiretroviral treatment (ART). ART was further defined as 1) non-highly active antiretroviral therapy (non-HAART) which included women not taking any ART as well as those taking a single, dual, or a combination of agents felt to be less potent based on current standards; 2) HAART with a protease inhibitor; and 3) HAART without a protease inhibitor.

For the purpose of this study, 25(OH)D levels were defined as sufficient if ≥32 ng/mL, 80 nmol/L. This cut point was selected based on prior research on optimal calcium homeostasis and bone health, recent clinical studies among pregnant women, and clinical applicability. [24,25] Additionally, data were divided into tertiles (rounded to the nearest 1 ng/ml) with 32 ng/mL defining the highest tertile. Sufficiency cut-points for $1,25(OH)D_2$ are less well established, therefore, tertiles (rounded to the nearest 1 pg/ml) based on the frequency distribution

were used in analyses. Both $1,25(OH)D_2$ and $25(OH)D$ were also evaluated as continuous variables. Variables associated with vitamin D levels were identified using linear regression models with either $25(OH)D$ or $1,25(OH)D_2$ as the dependent variable and demographic and clinical characteristics as the independent variables. Generalized estimating equations (GEE) were used to account for the correlation among mothers with multiple birth outcomes. Correlates of $25(OH)D$ and $1,25(OH)D_2$ were analyzed separately.

Univariate analyses of CMV infection were conducted using GEE logistic regression models with infant CMV status (positive/negative) as the outcome variable. Levels of $25(OH)D$, $1,25(OH)D_2$ and demographic and clinical characteristics were tested for their association with congenital CMV and peri/postnatal CMV in separate models

GEE logistic regression models were used in multivariate modeling of CMV. Separate models for $25(OH)D$ and $1,25(OH)D_2$ were analyzed for those with congenital CMV test results but only $1,25(OH)D_2$ for those with peri/postnatal results as no univariate relationship between $25(OH)D$ and peri/postnatal CMV infection was found. Factors associated with vitamin D and/or CMV in univariate models with *P-value* <0.15 were initially included in the multivariate models. Backward elimination was used to remove variables not associated with CMV (*P*>0.05). Final multivariate models controlled for CD4 count and HIV viral load due to their presumed impact on both vitamin D levels and CMV transmission. In the congenital group, covariates analyzed included ART category, race, ethnicity, HIV viral load, CD4 count, CD4 nadir during pregnancy, season, and maternal age at collection. In the peri/postnatal group, the covariates included ART category, race, ethnicity, HIV viral load, CD4 count, season, and mode of delivery (vaginal versus caesarian birth).

### Ethics statement

This study was approved by the University of Southern California's Office for the Protection of Research Subjects, Health Sciences IRB. This was a retrospective study using data from an ongoing, IRB approved Natural History Study, for which patients had signed consent. The current study posed no more than minimal risk and need for consent was waived.

## Results

Among the entire cohort of 340 women/infant pairs, 38 infants had a positive CMV test between the age of birth and 6 months (11%). All positive infant test results are listed in S1 Table. Among the 300 infants with CMV testing between birth and 3 weeks, 14 or 4.7% were congenitally infected. Among the 315 infants with a CMV test between 22 days and 6 months (excluding the congenital cases), 24 infants or 7.6% were peri/postnatally infected. Among these 24 infants, 54% had a first positive test between age 4 and 10 weeks of life, 8% between weeks 11–17, and 38% between 18 and 26 weeks. Only 1 of these 24 infants was not tested in the congenital period while the other 23 had at least 1 negative test during the congenital period.

Study cohort characteristics are summarized in Table 1 and are categorized based on infant CMV transmission category. The majority of women in the cohort self-identified as Hispanic white (73%) and were between the ages of 20 and 34 years. Most (80%) had HIV viral loads <400 copies/ml, 40% had CD4 counts above 500 cells/mm³, and 78.5% were on HAART regimens.

Maternal vitamin D levels are described in Table 1 and Fig 2A and 2B. Overall, two-thirds of the women had low vitamin D with $25(OH)D$ levels <32 ng/ml. Forty-two percent had insufficient $25(OH)D$ levels between 21–31 ng/ml, 24% had inadequate levels of 11–20 ng/ml,

**Table 1. Study cohort characteristics.**

| Variables | Total sample (N = 340) | Congenital CMV+ (N = 14) | Peri/postnatal CMV+ (N = 24) | CMV negative (N = 302) |
|---|---|---|---|---|
| Maternal age (years) | | | | |
| <20 | 30 (8.8%) | 2 (14.3%) | 2 (8.3%) | 26 (8.6%) |
| 20–34 | 240 (70.6%) | 12 (85.7%) | 17 (70.8%) | 211 (69.9%) |
| ≥ 35 | 70 (20.6%) | 0 | 5 (20.8%) | 65 (21.5%) |
| Race | | | | |
| Hispanic white | 248 (72.9%) | 8 (57.1%) | 22 (91.7%) | 218 (72.2%) |
| Non-Hispanic white | 11 (3.2%) | 1 (7.1%) | 0 | 10 (3.3%) |
| Black | 77 (22.7%) | 5 (35.7%) | 0 | 72 (23.8%) |
| Other | 4 (1.2%) | 0 | 2 (8.3%) | 2 (0.7%) |
| Plasma HIV-1 RNA (copies/mL) | | | | |
| <400 | 266 (80.1%) | 11 (84.6%) | 21 (95.5%) | 234 (78.8%) |
| 400+ | 66 (19.9%) | 2 (15.4%) | 1 (4.5%) | 63 (21.2%) |
| Data missing | 8 | 1 | 2 | 5 |
| CD4 cell count (cells/mm$^3$) | | | | |
| <200 | 39 (11.6%) | 3 (21.4%) | 3 (12.5%) | 33 (11.0%) |
| 200–500 | 163 (48.4%) | 4 (28.6%) | 10 (41.7%) | 149 (49.8%) |
| >500 | 135 (40.1%) | 7 (50.0%) | 11 (45.8%) | 117 (39.1%) |
| Data missing | 3 | 0 | 0 | 3 |
| ART regimen | | | | |
| non-HAART* | 74 (22%) | 2 (14.3%) | 4 (16.7%) | 68 (22.8%) |
| HAART without PI | 77 (22.9%) | 5 (35.7%) | 2 (8.3%) | 70 (23.5%) |
| HAART with PI | 185 (55.6%) | 7 (50%) | 18 (75%) | 160 (53.7%) |
| Data missing | 4 | 0 | 0 | 4 |
| Mode of delivery | | | | |
| Cesarean section | 137 (40.9%) | 4 (28.6%) | 5 (21.7%) | 128 (42.9%) |
| Vaginal | 198 (59.1%) | 10 (71.4%) | 18 (78.3%) | 170 (57.1%) |
| Data missing | 5 | 0 | 1 | 4 |
| Infant gender | | | | |
| Female | 155 (45.6%) | 8 (57.1%) | 13 (54.2%) | 134 (44.4%) |
| Male | 185 (54.4%) | 6 (42.9%) | 11 (45.8%) | 168 (55.6%) |
| Season | | | | |
| Summer/Fall | 163 (47.9%) | 5 (35.7%) | 10 (41.7%) | 148 (49%) |
| Winter/Spring | 177 (52.1%) | 9 (64.3%) | 14 (58.3%) | 154 (51%) |
| 25(OH)D levels, ng/ml | | | | |
| <23 | 111 (32.7%) | 3 (21.4%) | 7 (29.2%) | 101 (33.4%) |
| 23–31 | 117 (34.4%) | 9 (64.3%) | 7 (29.2%) | 101 (33.4%) |
| ≥ 32 | 112 (32.9%) | 2 (14.3%) | 10 (41.7%) | 100 (33.1%) |
| 1,25 (OH)D$_2$ pg/ml | | | | |
| median (IQR) | 95.6 (69.8–125.4) | 66.8 (54.9–73) | 67.9 (57.9–80.5) | 100 (73.5–128.9) |
| ≤ 74 | 116 (34.1%) | 11 (78.6%) | 17 (70.8%) | 78 (25.8%) |
| 75–115 | 118 (34.7%) | 2 (14.3%) | 4 (16.7%) | 112 (37.1%) |
| 116+ | 106 (31.2%) | 1 (7.1%) | 3 (12.5%) | 112 (37.1%) |

CD4, and HIV RNA levels tested at time point closest available to the vitamin D specimen test date

*non-HAART includes women on no ART, single agents, dual combinations, or any ART felt to be less active compared to current standards

Abbreviations: ART, antiretroviral therapy; HAART, highly active antiretroviral therapy; IQR, Interquartile range; PI, protease-inhibitor; 25(OH)D, 25-hydroxyvitamin D; 1,25 (OH)D$_2$, 1,25-dihydroxyvitamin D

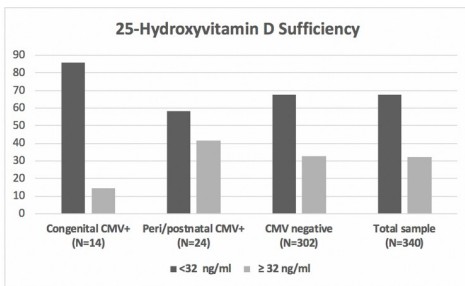
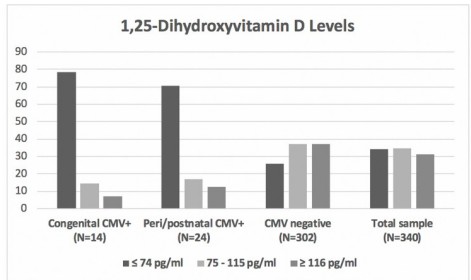

**Fig 2. a.** The percentage of mothers with 25(OH)D levels above and below the cut point of 32 ng/ml compared by the CMV status of their infants. **b.** The percentage of mothers with low, middle or high 1,25-dihydroxyvitamin D levels, based on tertiles, compared by CMV transmission status.

and 2% were deficient with levels of $\leq 10$ ng/ml (S2 Table). Maternal 25(OH)D and 1,25(OH)$D_2$ levels were only weakly correlated (Spearman's correlation coefficient = 0.21, $P = 0.0001$).

Table 2 summarizes the variables associated with 25(OH)D and 1,25(OH)$D_2$. Maternal HIV viral load was significantly associated with both 25(OH)D ($P = 0.002$) and 1,25(OH)$D_2$ ($P = 0.02$) with lower levels seen in women with higher viral loads. Lower 25(OH)D levels were associated with lower maternal CD4 cell counts ($P = 0.05$). Maternal ART category was significantly associated with 25(OH)D levels ($P = 0.001$) with the lowest values seen in the group on either no medications or non-HAART regimens. Season was significantly associated

**Table 2. Factors univariately associated with vitamin D levels.**

| | 25(OH)D ng/ml | | 1,25 (OH)D$_2$ pg/ml | |
|---|---|---|---|---|
| **Variables** | **Mean (95%CI)**[*] | ***P*-value** | **Mean (95% CI)**[*] | ***P*-value** |
| **ART regimen** | | 0.001 | | 0.07 |
| non-HAART | 24.1 (21.6–26.6) | | 89.9 (80.5–99.4) | |
| HAART no PI | 28.8 (26.6–30.9) | | 104.5 (95.0–113.9) | |
| HAART with PI | 31.1 (29.3–32.9) | | 101.0 (95.5–106.6) | |
| **Race** | | 0.13 | | 0.35 |
| Hispanic white | 28.9 (27.3–30.5) | | 101.3 (95.7–106.9) | |
| Non-Hispanic white | 36.7 (28.4–44.9) | | 101.7 (77.8–125.6) | |
| Black | 27.4 (24.7–30.1) | | 92.6 (84.5–100.8) | |
| Other | 36.4 (28.3–44.6) | | 84.6 (41.7–127.6) | |
| **Plasma HIV-1 RNA (copies/mL)** | | 0.002 | | 0.02 |
| <400 | 30.1 (28.7–31.6) | | 102.5 (97.5–107.4) | |
| 400+ | 24.8 (22.0–27.6) | | 89.1 (80.4–97.8) | |
| **CD4 cell count (cells/mm$^3$)** | | 0.05 | | 0.07 |
| $\geq 200$ | 29.4 (27.9–30.9) | | 100.1 (95.3–104.9) | |
| <200 | 25.1 (21.4–28.8) | | 88.0 (76.5–99.5) | |
| **Season** | | 0.007 | | 0.002 |
| Summer & Fall | 30.9 (28.7–33.0) | | 105.9 (99.3–112.6) | |
| Winter & Spring | 27.2 (25.6–28.8) | | 92.8 (87.0–98.6) | |

GEE linear regression model (separate models for 25(OH)D and 1,25 (OH)$D_2$)

CD4, and HIV RNA levels tested at time point closest available to the vitamin D specimen test date

[*]Least squares mean with 95% confidence interval

Abbreviations: ART, antiretroviral therapy; HAART, highly active antiretroviral therapy; PI, protease inhibitor; 25 (OH)D, 25-hydroxyvitamin D; 1,25(OH)$D_2$, 1,25-dihydroxyvitamin D

with both 25(OH)D ($P = 0.007$) and 1,25(OH)$D_2$ ($P = 0.002$), with lower levels among women pregnant during months with lower levels of UV light. No relationships were found between CD8 cell counts nor CD4/CD8 ratios and either 25(OH)D or 1,25(OH)$D_2$.

Factors associated with congenital transmission of CMV are summarized in Tables 3, 4 and 5. Lower levels of maternal bioactive vitamin D, 1,25(OH)$D_2$, were associated with increased congenital transmission of CMV in univariate ($P = 0.006$) and multivariate analyses ($P = 0.009$ for trend) controlling for maternal age, CD4 count, and HIV viral load categories. Women with 1,25(OH)$D_2$ levels ≤74 pg/ml were 12 times more likely to transmit CMV congenitally to their infant (OR 12.2 [95% CI 1.61–92.2] $P = 0.02$) compared to women with levels in the highest tertile of ≥116 pg/ml.

In univariate analyses, women with 25(OH)D levels in the middle range of 23–31 ng/dl were more likely to transmit congenital CMV (OR 4.93 [1.05–23.1] $P = 0.04$) compared to women with levels ≥32. Borderline significance for this association was observed in the multivariate analysis when controlling for maternal age, CD4 count, and HIV viral load categories (OR 4.88 [95% CI 0.99–24.1] $P = 0.05$). However, there was no significant difference found when comparing the lowest and highest tertile groups, nor when a cut point for vitamin D sufficiency of 32 ng/ml was used. Younger maternal age was also associated with increased risk of congenital CMV infection in univariate analysis (OR 0.90 [95% CI 0.83–0.98] $P = 0.01$) and remained significant in the multivariate model (OR 0.90 [95% CI 0.83–0.99] $P = 0.03$).

Factors associated with peri/postnatal CMV infection status in both univariate and multivariate analyses are summarized in Tables 3 and 6. In univariate analyses, only 1,25(OH)$D_2$ was associated with peri/postnatal CMV infection ($P = 0.0002$). In addition, vaginal delivery compared to cesarean section was marginally associated with increased odds of peri/postnatal transmission. However, in multivariate analyses controlling for HIV viral load categories, CD4 count, 1,25(OH)$D_2$ and ART, vaginal deliveries had 3 times greater odds of peri/postnatal transmission (OR 3.03 [95% CI 1.08–8.50] $P = 0.04$). Levels of 1,25(OH)$D_2$ remained significant in multivariate analyses with higher maternal 1,25(OH)$D_2$ levels protective for peri/postpartum CMV. Women with levels ≤ 74 pg/ml had nearly 10 times greater odds peri/postpartum transmission compared to women with 1,25(OH)$D_2$ levels ≥ 116 pg/ml controlling for mode of delivery, CD4 and HIV viral load categories, and ART (OR 9.84 [95% CI 2.63–36.8] $P = 0.003$). There was no significant association seen between maternal 25(OH)D status and peri/postnatal CMV.

## Discussion

This is the first study to evaluate the impact of maternal vitamin D status on congenital and peri/postnatal CMV transmission. In this study, lower levels of the bioactive form of vitamin D, calcitriol (1,25(OH)$D_2$), were associated with increased odds of both congenital and peri/postpartum CMV infections among perinatally exposed but HIV uninfected infants born to non-breastfeeding women with HIV. As seen in other cohorts, low levels of vitamin D were associated with lower maternal CD4 counts and higher HIV viral loads.[26–29] This supports the notion that vitamin D is involved in the immune functions related to HIV infection. However, the relationships found between calcitriol and CMV transmission outcomes in this study were independent of these factors.

There is growing evidence suggesting the importance of vitamin D in placental-related functions including placental development and implantation, calcium transport, immuno-modulatory functions, as well as an association between low calcitriol levels and pregnancy induced hypertension. [15,30–33] During pregnancy, non-classical, extra-renal production of 1,25(OH)$D_2$ occurs in the placenta. In fact, healthy pregnancy is associated with a doubling or

**Table 3. Factors univariately associated with congenital and perinatal/postnatal CMV transmission.**

| Variable | Congenital CMV | | Perinatal/Postnatal CMV | |
|---|---|---|---|---|
| | OR (95% CI) | *P*-value | OR (95% CI) | *P*-value |
| **25(OH)D* (ng/mL)** | 0.99 (0.94–1.04) | 0.59 | 1.01 (0.99–1.04) | 0.35 |
| **25(OH)D (ng/mL)** | | 0.07 | | 0.61 |
| ≥ 32 | 1.00 | | 1.00 | |
| 23–31 | 4.93 (1.05–23.1) | **0.04** | 0.65 (0.24–1.78) | 0.40 |
| <23 | 1.69 (0.28–10.3) | 0.57 | 0.66 (0.24–1.78) | 0.41 |
| **1,25(OH)D$_2$* (pg/mL)** | 0.97 (0.95–0.98) | **0.0002** | 0.98 (0.97–0.99) | **0.001** |
| **1,25(OH)D$_2$ (pg/mL)** | | **0.006** | | **0.0002** |
| ≥ 116 | 1.00 | | 1.00 | |
| 75–115 | 2.01 (0.18–22.3) | 0.57 | 1.32 (0.29–6.01) | 0.72 |
| ≤ 74 | 12.6 (1.66–99.5) | 0.01 | 7.88 (2.23–27.8) | **0.001** |
| **ART Regimen** | | 0.46 | | 0.11 |
| Non-HAART | 1.00 | | 1.00 | |
| HAART without PI | 2.29 (0.43–12.2) | 0.33 | 0.49 (0.09–2.72) | 0.41 |
| HAART with PI | 1.18 (0.24–5.82) | 0.84 | 2.03 (0.67–6.17) | 0.21 |
| **Mode of delivery** | | | | |
| C-section | 1.00 | | 1.00 | |
| Vaginal | 1.79 (0.55–5.80) | 0.33 | 2.67 (0.97–7.34) | 0.06 |
| **Baby gender** | | | | |
| Male | 1.00 | | 1.00 | |
| Female | 1.66 (0.56–4.90) | 0.36 | 1.55 (0.67–3.58) | 0.30 |
| **Maternal Age** | 0.90 (0.83–0.98) | **0.01** | 0.99 (0.94–1.05) | 0.81 |
| **Race** | | | ** | |
| Non-Black | 1.00 | | | |
| Black | 2.03 (0.67–6.17) | 0.21 | | |
| **Ethnicity** | | | | |
| Non-Hispanic | 1.00 | | 1.00 | |
| Hispanic | 0.42 (0.14–1.23) | 0.11 | 8.43 (1.12–63.5) | **0.04** |
| **Plasma HIV-1 RNA (copies/mL)** | | | | |
| <400 | 1.00 | | 1.00 | |
| 400+ | 0.80 (0.17–3.74) | 0.78 | 0.18 (0.02–1.28) | 0.09 |
| **CD4 Nadir During Pregnancy** | | | | |
| 200+ | 1.00 | | 1.00 | |
| <200 | 1.72 (0.52–5.71) | 0.37 | 1.03 (0.37–2.88) | 0.95 |
| **CD4 Cell Count (copies/mL)** | | | | |
| 200+ | 1.00 | | 1.00 | |
| <200 | 2.31 (0.61–8.74) | 0.22 | 1.08 (0.30–3.90) | 0.90 |
| **Season** | | | | |
| Summer/Fall | 1.00 | | 1.00 | |
| Winter/Spring | 1.72 (0.57–5.22) | 0.34 | 1.31 (0.56–3.04) | 0.53 |

* Modeled as a continuous variable

**Race was not included in the perinatal/postnatal model because no Black infants were perinatal/postnatal CMV+

GEE logistic regression model

Abbreviations: ART, antiretroviral therapy; CI, confidence interval; HAART, highly active antiretroviral therapy; OR, odds ratio; PI, protease inhibitor; 1,25(OH)D2, 1,25-dihydroxyvitamin D; 25(OH)D, 25-hydroxyvitamin D

**Table 4. Factors associated with congenital CMV in multivariate model A.**

| Variables | OR (95% CI) | *P*-value | *P*-value* |
|---|---|---|---|
| **CMV** | | | |
| **1,25(OH)D$_2$ (pg/mL)** | | **0.01** | **0.009** |
| $\geq$ 116 | 1.00 | | |
| 75–115 | 2.18 (0.20–23.6) | 0.52 | |
| $\leq$ 74 | 12.2 (1.61–92.2) | **0.02** | |
| **Plasma HIV-1 RNA (copies/mL)** | | | |
| <400 | 1.00 | | |
| 400+ | 0.41 (0.08–2.00) | 0.27 | |
| **CD4 Cell Count (copies/mL)** | | | |
| 200+ | 1.00 | | |
| <200 | 2.07 (0.42–10.3) | 0.37 | |
| **Maternal age** | 0.90 (0.83–0.99) | **0.03** | |

*\*P*-value for trend

GEE logistic regression model

Abbreviations: CI, confidence interval; OR, odds ratio; 1,25(OH)D2, 1,25-dihydroxyvitamin D

tripling of pre-pregnancy 1,25(OH)D$_2$ levels while 25(OH)D levels typically remain unchanged.[25,34] This increase occurs without impacting serum or urinary calcium levels, demonstrating an uncoupling of vitamin D metabolism from the usual calcium-parathyroid hormone axis control.[33,34] Placental trophoblasts and maternal decidua actively convert vitamin D to its active form, 1,25(OH)D$_2$, in an intracrine manner.[32] Decidual and placental cells also contain an abundance of the vitamin D receptor (VDR), a ligand-activated transcription factor that controls the expression of over a thousand genes, allowing for a localized response when 1,25(OH)D$_2$ binds. This response involves activation of the innate immune system including a dose-dependent production of the antimicrobial peptide, cathelicidin, ultimately protecting placental cells from infection and death.[18,35] It can be postulated that the

**Table 5. Factors associated with congenital CMV in multivariate model B.**

| Variables | OR (95% CI) | *P*-value | *P*-value* |
|---|---|---|---|
| **CMV** | | | |
| **25(OH)D (ng/mL)** | | 0.10 | 0.46 |
| $\geq$ 32 | 1.00 | | |
| 23–31 | 4.88 (0.99–24.1) | **0.05** | |
| <23 | 1.87 (0.27–12.9) | 0.53 | |
| **Plasma HIV-1 RNA (copies/mL)** | | | |
| <400 | 1.00 | | |
| 400+ | 0.42 (0.06–3.04) | 0.39 | |
| **CD4 Cell Count (copies/mL)** | | | |
| 200+ | 1.00 | | |
| <200 | 3.08 (0.53–18.0) | 0.21 | |
| **Maternal age** | 0.90 (0.82–0.98) | **0.02** | |

*\*P*-value for trend

GEE logistic regression model

Abbreviations: CI, confidence interval; OR, odds ratio; 25(OH)D, 25-hydroxyvitamin D

**Table 6. Factors associated with peri/postnatal CMV in multivariate model.**

| Variables | CMV | | P-value* |
| --- | --- | --- | --- |
| | OR (95% CI) | P-value | |
| **1,25(OH)D$_2$ (pg/mL)** | | **0.0003** | **0.0008** |
| $\geq$ 116 | 1.00 | | |
| 75–115 | 1.60 (0.34–7.50) | 0.55 | |
| $\leq$ 74 | 9.84 (2.63–36.8) | **0.0007** | |
| **Plasma HIV-1 RNA (copies/mL)** | | | |
| <400 | 1.00 | | |
| 400+ | 2.18 (0.65–7.32) | 0.21 | |
| **CD4 Cell Count (copies/mL)** | | | |
| 200+ | 1.00 | | |
| <200 | 1.25 (0.57–2.76) | 0.58 | |
| **ART Regimen** | | **0.05** | |
| Non-HAART | 1.00 | | |
| HAART no PI | 0.83 (0.10–7.25) | 0.87 | |
| HAART with PI | 4.21 (0.75–23.8) | 0.10 | |
| **Mode of delivery** | | | |
| C-section | 1.00 | | |
| Vaginal | 3.03 (1.08–8.50) | **0.04** | |

*P-value for trend

GEE logistic regression model

Abbreviations: ART, antiretroviral therapy; CI, confidence interval; HAART, highly active antiretroviral therapy; OR, odds ratio; PI, protease inhibitor; 1,25(OH)D2, 1,25-dihydroxyvitamin D; 25(OH)D, 25-hydroxyvitamin D

additional 1,25(OH)D$_2$ available to the developing fetus is in excess of the amount required for fetal skeletal development and instead may be essential for fetal and placental immunologic and antimicrobial functions. Our findings support this notion suggesting that the increased 1,25(OH)D$_2$ available during pregnancy could aid immune functions important in blocking transplacental CMV transmission and possibly decrease maternal CMV shedding, thereby reducing congenital and peri/postnatal CMV infections.

A prior study of untreated HIV infected pregnant women in Tanzania demonstrated increased mother-to-child-transmission of HIV in women with 25(OH)D levels <32 ng/ml. [24] In the current study, we found only weak evidence that lower levels of 25(OH)D, the main circulating form of vitamin D, may be associated with congenital CMV transmission. There was no trend as only women with levels in the middle tertile appeared to have increased CMV transmission compared to those with the highest levels, and there was no difference when comparing those with the lowest and highest levels, nor when the cut point for vitamin D sufficiency of 32 ng/ml was used for analysis. A randomized controlled trial of vitamin D supplementation in HIV negative pregnant women by Hollis, et al, demonstrated that a serum 25(OH)D level of 40 ng/ml was needed in order to achieve the supraphysiological increase in the bioactive form of vitamin D, calcitriol (1,25(OH)D$_2$), during normal pregnancy.[25] It is possible a larger sample size is needed to similarly find a threshold serum value of 25(OH)D required to overcome any problems with placental dysfunction, supporting the necessary boost in calcitriol, and thereby aiding the important immunologic protective functions of the placenta.

This study had several limitations. This is a retrospective study and the cohort was created, in part, based on availability of stored samples. This may have created inadvertent bias in

cohort selection. The women in this cohort were all considered seropositive for CMV based on maternal IgG, or early infant IgG in a few cases. Therefore, it can be assumed that the transmissions during the congenital period were all non-primary infections due to reactivation of latent maternal CMV, or theoretically, reinfection with a different strain. Further study is needed to determine if vitamin D plays a role in low-seroprevalence populations as well, where higher rates of maternal primary CMV infection are more likely. The retrospective study design also limited the ability to do additional or confirmatory CMV PCR testing on infant urine, saliva or blood, due to the lack of availability of stored specimens. The results of this study were based mainly on culture results, as only 55 of the 340 infants had blood PCR test results, and there were no PCR results for urine or saliva. CMV cultures are less sensitive than PCR for detecting CMV and therefore the actual prevalence of congenital and peri/postnatal CMV infections may be underrepresented. More work is needed to demonstrate if the association between calcitriol, the active form of vitamin D, and congenital and early postnatal transmission of CMV found in this population of HIV-infected women is true in HIV uninfected women as well. Additionally, this study did not evaluate VDR expression nor the multitude of polymorphisms associated with altered VDR expression and function. It is unclear how the VDR and therefore the downstream effects of vitamin D may be impacted by CMV infection itself. In fact, a recent in vitro study demonstrated that the presence of CMV inhibited the expression of vitamin D receptors in fibroblasts.[36] Further study is needed to clarify the interplay between vitamin D and CMV infection and how this relationship may be important in protecting the developing fetus from potential infection.

Although our study demonstrates an association between lower calcitriol $(1,25(OH)D_2)$ levels and increased CMV transmission congenitally and in the early postnatal period, we cannot comment on causation. It is unknown if the CMV and/or HIV infections deplete vitamin D as it is used in high demand in conjunction with fighting these infections, or if low maternal vitamin D leads to increased susceptibility to infection and/or increased or prolonged viral shedding. HIV and CMV infections are both known to cause significant placental damage and dysfunction and this could impact the placental production of calcitriol $(1,25(OH)D_2)$, potentially contributing to the lower levels seen in this study.[37–39] Additionally, the presence of pathogens may affect local calcitriol production. In vitro, the presence of HIV and LPS impacts the production and breakdown of calcitriol by affecting CYP27B1 and CYP24A1 gene expression in monocytes.[40]

Future, prospective studies are needed in HIV positive and negative populations to further clarify these complex relationships. If found to be causal and protective in prospective studies, targeted vitamin D supplementation, with the goal of supporting the necessary rise in calcitriol during pregnancy, could represent a safe and inexpensive tool in preventing CMV transmission from mother to infant.

## Supporting information

**S1 Table. Number of positive CMV tests by test type for all CMV+ infants.**
(DOCX)

**S2 Table. 25-Hydroxyvitamin D sufficiency by infant transmission category.**
(DOCX)

## Acknowledgments

The authors would like to acknowledge the MCA Laboratory staff for their assistance with this study.

## Author Contributions

**Conceptualization:** Allison Bearden, Kristi Van Winden.

**Data curation:** Allison Bearden, Toni Frederick, Naoko Kono, Lorayne Barton.

**Formal analysis:** Toni Frederick, Naoko Kono.

**Funding acquisition:** Allison Bearden, Kristi Van Winden.

**Investigation:** Allison Bearden, Kristi Van Winden, Raj Pandian.

**Methodology:** Allison Bearden, Kristi Van Winden, Toni Frederick, Naoko Kono, Eva Operskalski, Andrea Kovacs.

**Project administration:** Allison Bearden.

**Resources:** Andrea Kovacs.

**Supervision:** Allison Bearden, Alice Stek, Andrea Kovacs.

**Writing – original draft:** Allison Bearden.

**Writing – review & editing:** Allison Bearden, Toni Frederick, Naoko Kono, Eva Operskalski, Alice Stek, Andrea Kovacs.

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
