## [Decision Letter · Decision Letter 0]

8 Nov 2019

PONE-D-19-28840

Low Maternal Vitamin D is Associated with Increased Risk of Congenital and Peri/postnatal Transmission of Cytomegalovirus in Women with HIV

PLOS ONE

Dear Dr. Bearden,

Thank you for submitting your manuscript to PLOS ONE. After careful consideration, we feel that it has merit but does not fully meet PLOS ONE’s publication criteria as it currently stands. Therefore, we invite you to submit a revised version of the manuscript that addresses the points raised by the expert who reviewed your study.

Could you please further discuss why you did not screen urine/saliva with PCR which would have been more sensitive? Please also include a breakdown of the samples that tested positive as this may indicate the level of replication.

We would appreciate receiving your revised manuscript by Dec 23 2019 11:59PM. To enhance the reproducibility of your results, we recommend that if applicable you deposit your laboratory protocols in protocols.io, where a protocol can be assigned its own identifier (DOI) such that it can be cited independently in the future. For instructions see: http://journals.plos.org/plosone/s/submission-guidelines#loc-laboratory-protocols

We look forward to receiving your revised manuscript.

Kind regards,

Michael Nevels, Ph.D.

Academic Editor

PLOS ONE

Journal Requirements:

2. We ask that you please include in your methods section, the full ethics statement from your on-line details.

Reviewers' comments:

Reviewer's Responses to Questions

**Comments to the Author**

1. Is the manuscript technically sound, and do the data support the conclusions?

Reviewer #1: Yes

2. Has the statistical analysis been performed appropriately and rigorously? 

Reviewer #1: Yes

3. Have the authors made all data underlying the findings in their manuscript fully available?

Reviewer #1: Yes

4. Is the manuscript presented in an intelligible fashion and written in standard English?

Reviewer #1: Yes

5. Review Comments to the Author

Reviewer #1: This is an interesting study which appears to identify low levels of vitamin D being associated with risk of congenital and peri/postnatal CMV infection. The study has many limitations which the authors acknowledge in their discussion. However, the statistical analysis is performed very comprehensively and the data support a role for vitamin D in this cohort.

I would like the authors to comment on/correct the following areas prior to acceptance:

1) it is not clear for the neonates testing positive how many were serum positive, urine positive or saliva positive. These data should be included perhaps as a Venn diagram (supplementary figure) where the reader can see the overlap between positivity in the different sites.

2) It is disappointing that PCR was not used for the analysis of urine and saliva samples. Whilst congenital infants often have high levels of CMV in their urine and can be easily cultured I wonder whether the authors might have missed some neonates especially in the peri/postnatal period due to the inherent sensitivity of culture methods. Would it be possible for the authors to perform this analysis or indeed expand the study if new infants were identified through PCR of urine/saliva?

3) The study cohort traverses 14 years. Do you authors think that storage might have had an influence on any of the measurements. Perhaps a timeline of vitamin D levels with recruitment date might be useful to ensure this has not introduced any further bias.

6. PLOS authors have the option to publish the peer review history of their article (what does this mean?). If published, this will include your full peer review and any attached files.

Reviewer #1: No

---

## [Author Response · Author response to Decision Letter 0]

14 Jan 2020

1. Rationale for not screening the infant urine and saliva samples for CMV by PCR: 

This study was a retrospective study nested in a prospective natural history study. The infant samples were originally tested based on the protocol for the natural history study which entailed only CMV cultures. Unfortunately, there were not sufficient samples available to retest for CMV using PCR methodology. This may have underestimated the prevalence of CMV infections in the infants. This explanation has been added to the discussion section, in the paragraph describing study limitations. 

2. Inclusion of a breakdown of samples which tested positive:

In order to provide additional information regarding which tests were positive for infants categorized as CMV infected, a table was added as supplementary data, and cited in the manuscript in the first paragraph of the results section (S1 Table. Number of positive CMV tests by test type for all CMV+ infants)

3. Style requirements: 

Requirements have been reviewed and the revised document has incorporated all necessary changes. 

4. Ethics statement: 

The ethics statement from the online submission was added to the manuscript at the end of the Methods section. 

5. Data accessibility: 

We are unable to publicly share the dataset from this study due to ethical and legal restrictions. According the University of Southern California’s Office of Ethics and Compliance, there is no way to make the dataset truly de-identified. The dataset is made up solely of women with a diagnosis of HIV and their offspring. The diagnosis of HIV is an indirect identifier and considered sensitive information requiring additional protections. Additional indirect identifiers in the dataset include age, race/ethnicity, and sex. Additionally, the consent signed by the study participants did not include a provision for their data to be made publicly accessible. Data requests can be directed to department administrator, Carlota Obnillas, obnillas@usc.edu, and if the request is approved, the USC Stevens Center for Innovation will assist and create a Data Transfer Agreement.

6. Data not shown:

Data to support the statements summarizing the 25-hydroxyvitamin D levels of the cohort, in the third paragraph of the results section, has been added as a supplementary table (S2 Table 25-Hydroxyvitamin D sufficiency by infant transmission category)

7. Concern regarding the influence of storage time on vitamin D measurements:

Vitamin D is established as very stable in frozen specimens and can be stored indefinitely according to several lab manuals or processing instructions. The levels also maintain stability in the face of multiple repeated freeze-thaw cycles as well. This information has been added to the manuscript in the methods section, in the paragraph on Vitamin D testing. 

https://www.nationaljewish.org/for-professionals/diagnostic-testing/adx/tests/vitamin-d-25-hydroxy-total

---

## [Editor Report · Decision Letter 1]

27 Jan 2020

Low maternal vitamin D is associated with increased risk of congenital and peri/postnatal transmission of Cytomegalovirus in women with HIV

PONE-D-19-28840R1

Dear Dr. Bearden,

We are pleased to inform you that your manuscript has been judged scientifically suitable for publication and will be formally accepted for publication once it complies with all outstanding technical requirements.

With kind regards,

Michael Nevels

Academic Editor

PLOS ONE
---

## [Editor Report · Acceptance letter]

31 Jan 2020

PONE-D-19-28840R1 

Low maternal vitamin D is associated with increased risk of congenital and peri/postnatal transmission of Cytomegalovirus in women with HIV 

Dear Dr. Bearden:

I am pleased to inform you that your manuscript has been deemed suitable for publication in PLOS ONE. Congratulations! Your manuscript is now with our production department. 

With kind regards,

on behalf of

Dr. Michael Nevels 

Academic Editor

PLOS ONE